# Encapsulation of a *Desmodium intortum* Protein Isolate Pickering Emulsion of β-Carotene: Stability, Bioaccesibility and Cytotoxicity

**DOI:** 10.3390/foods11070936

**Published:** 2022-03-24

**Authors:** Xue-Mei Tang, Pan-Dao Liu, Zhi-Jian Chen, Xin-Yong Li, Rui Huang, Guo-Dao Liu, Rong-Shu Dong, Jian Chen

**Affiliations:** 1Institute of Tropical Crop Genetic Resources, Chinese Academy of Tropical Agriculture Sciences, Haikou 571101, China; tangxm1996@163.com (X.-M.T.); liupd2018@163.com (P.-D.L.); jchen@scau.edu.cn (Z.-J.C.); lixy05@163.com (X.-Y.L.); bluesing@126.com (R.H.); liuguodao2008@163.com (G.-D.L.); 2Key Laboratory of Food Nutrition and Functional Food of Hainan Province, One Health Institute, Engineering Research Center of Utilization of Tropical Polysaccharide Resources, Ministry of Education, College of Food Science and Technology, Hainan University, Haikou 570228, China

**Keywords:** Pickering emulsion, β-carotene, simulated in vitro digestion, bioaccessibility, cytotoxicity

## Abstract

Owing to their excellent characteristics, Pickering emulsions have been widely used in the development and the application of new carriers for embedding and for delivering active compounds. In this study, β-carotene was successfully encapsulated in a Pickering emulsion stabilized using *Desmodium intortum* protein isolate (DIPI). The results showed that the encapsulation efficiencies of β-carotene in the control group Tween 20 emulsion (TE) and the DIPI Pickering emulsion (DIPIPE) were 46.7 ± 2.5% and 97.3 ± 0.8%, respectively. After storage for 30 days at 25 °C and 37 °C in a dark environment, approximately 79.4% and 72.1% of β-carotene in DIPIPE were retained. Compared with TE, DIPIPE can improve the stability of β-carotene during storage. In vitro digestion experiments showed that the bioaccessibility rate of β-carotene in DIPIPE was less than that in TE. Cytotoxicity experiments showed that DIPI and β-carotene micelles within a specific concentration range exerted no toxic effects on 3T3 cells. These results indicate that DIPIPE can be used as a good food-grade carrier for embedding and transporting active substances to broaden the application of the protein-based Pickering emulsion system in the development of functional foods.

## 1. Introduction

A Pickering emulsion refers to an emulsion stabilized by solid particles instead of surfactants. The interface layer formed by solid particles ensures that the Pickering emulsions exert satisfactory embedding and carrying effects. Numerous studies have shown that a Pickering emulsion can be loaded with biologically active substances or drugs to complete delivery and release [1]. Moreover, a Pickering emulsion stabilized with biobased solid particles such as protein [2] and starch [3] provides several advantages, including high biological safety and good biocompatibility, and it is easy to prepare. It only needs to disperse the emulsifier in one phase to form a uniform dispersion system. The other phase is added to it and it can be prepared by simple mechanical high-speed stirring or ultrasonic rapid oscillation. Pickering emulsion has become one of the main objects of food functional factor carrier research [4]. As a carrier, an oil-in-water (O/W) Pickering emulsion can encapsulate lipophilic substances such as β-carotene [5] and curcumin [6] in the internal phase, as well as improve its stability and bioaccessibility. Tang et al. [7] used soy globulin nanoparticles to prepare a Pickering emulsion to embed β-carotene. The results show that the slow-release effect of β-carotene is mainly related to the size of the emulsion droplets. In the gelatinous emulsion stabilized by heat-treated glycinin, the formation of a gel network can greatly reduce the release rate of β-carotene. Marefati et al. [8] used a Pickering emulsion stabilized by starch granules to encapsulate curcumin, and they simulated the digestive stability of curcumin in the mouth, stomach, and small intestine. The experiment revealed that the emulsion successfully protected curcumin in the mouth and stomach, allowing curcumin to be absorbed in the small intestine.

A natural pigment with a conjugated polyene double bond structure, β-Carotene cannot be synthesized by the human body [9]. It has a variety of biological activities such as antioxidant functions [10], and the promotion of intercellular gap junction communication; and, it plays an important role in delaying the aging of the body, protecting cells from damage [11], and reducing the risk of cancer [12,13]. However, β-carotene is easily oxidized and degraded under high temperature and light and oxygen conditions, resulting in the loss of its biological activity [14]. In addition, the application of β-carotene in the food industry is limited due to its poor water solubility and low bioavailability. To solve the aforementioned problems, the embedded delivery system of β-carotene has often drawn research interest in the food sector. Relevant research shows that β-carotene can be embedded in emulsions, liposomes, solid lipid nanoparticles, and macromolecular complexes [15,16]. Among these carriers, emulsions are the most widely investigated. Fu et al. [5] encapsulated β-carotene in a wheat gluten protein Pickering emulsion to improve the bioavailability of β-carotene. Fang et al. [17] used octenyl succinate anhydride modified starch and chitosan to prepare a multilayer nanoemulsion; the complex interface layer of the emulsion further protected β-carotene.

A novel Pickering stabilizer was successfully developed in previous laboratory research that used a protein isolate from *Desmodium intortum* [18]. The DIPIPE had a gel-like network structure, and it still had high stability after 30 days of storage when prepared under the condition that the protein concentration was 5% (*w*/*v*) and the volume ratio of the protein water phase to the oil phase was 3:7.

In this study, DIPI Pickering emulsion was used to encapsulate β-carotene, and the traditional Tween 20 emulsion was used as a control to investigate the encapsulation and the retention of β-carotene in the Pickering emulsion. The encapsulated β-carotene digestibility was evaluated by in vitro simulated digestion experiments. In addition, there are no research reports on the toxicity of DIPI and its application. Therefore, in this study, the safety of DIPI and its stabilized β-carotene Pickering emulsion was confirmed by cytotoxicity experiments. These results provide a technical way to solve the limitation of β-carotene in the food industry, and they also provide a theoretical basis for the application of DIPIPE in the encapsulation and the delivery of active substances.

## 2. Materials and Methods

### 2.1. Materials

DIPI is made by the laboratory [18], the protein content in DIPI (N × 6.25) was 90.7 ± 0.46%, as determined by the Kjeldahl method (wet basis). Soybean oil was purchased from a local supermarket (Haikou, China) without further purification. The β-Carotene (≥96%), pepsin (≥2500 units/mg dry weight), and pancreatin (Protease activity: 100–350 U/mg, Amylase activity: 100–350 U/mg, Lipase activity: 10–75 U/mg) were purchased from Aladdin Biochemical Technology Co., Ltd. (Shanghai, China). The Cell Counting Kit-8 (CCK-8) was purchased from YEASEN Biological Technology Co., Ltd. (Shanghai, China). Bile salts were purchased from Macklin Biochemical Technology Co., Ltd. (Shanghai, China). All other reagents were of analytical grade.

### 2.2. Preparation of Emulsion-Encapsulated β-Carotene

The β-carotene crystal powder used had a melting point of 183 °C, which was only slightly soluble in oil, and it was very difficult to directly dissolve the crystal powder in oil at room temperature. The process of dissolving β-carotene in vegetable oil requires a long period of time and a temperature of 150 °C. In this study, in order to avoid the thermal degradation of β-carotene, ultrasonic pretreatment was used and then heated at 90 °C to dissolve β-carotene.

To adjust the β-carotene concentration to 0.2% (*w*/*v*), β-Carotene was added to soybean oil, sonicated for 1 min, and heated in a water bath at 90 °C for 10 min to ensure that the β-carotene was completely dissolved in the soybean oil. The most stable method of preparing DIPIPE obtained in the previous laboratory study was used as the reference for preparing the β-carotene Pickering emulsion [18]. Specifically, the DIPI solution (5%, *w*/*v*) was prepared, subjected to heat at 95 °C for 30 min, and NaCl was added to adjust the ionic strength to 150 mM. The protein water phase and the β-carotene oil phase were then mixed in a ratio of 3:7. Mixing was performed using the IKA-ULTRATURRAX T25 digital disperser (IKA 190 Works, Inc., Wilmington, NC, USA) for 2 min at 20,000 rpm. The β-carotene Pickering emulsion was then obtained; Tween 20 emulsion (TE) was used as the control group. The Tween 20 was dissolved in ultrapure water at a concentration of 2%, and the Tween 20 aqueous phase was mixed with the β-carotene oil phase in a ratio of 1:1 by using a disperser. An emulsion was then prepared. The β-carotene emulsion was prepared under dark conditions.

### 2.3. Extraction and Quantification of β-Carotene

To determine the β-carotene content, a standard curve was first generated. A β-carotene standard solution at a certain concentration (0–10 μg/mL), mixed with n-hexane, was prepared. Gradient dilution was then conducted, and the ultraviolet absorption value of the solution at 450 nm was measured using an ultraviolet-visible spectrophotometer (UV-5500PC spectrophotometer; Metash Instruments Co., Ltd., Shanghai, China). A standard curve was generated, with β-carotene concentration (μg/mL) as the abscissa and absorbance (A) as the ordinate.

Using Wright’s method, β-Carotene was extracted [19], with certain modifications. The process was as follows: up to 2 mL of absolute ethanol and 2 mL of n-hexane were added to 1 mL of the emulsion sample. The sample was oscillated on a vortex vortexer to fully demulsify and dissolve the sample and then allowed to stand for 10 min. The layers were separated, and the n-hexane phase was collected. The remaining samples were extracted twice with n-hexane until the n-hexane phase became less. The extraction was conducted under dark conditions. The extracts were combined and then diluted into a 25 mL brown volumetric flask, and the absorbance at 450 nm was measured for quantitative analysis.

### 2.4. Encapsulation Efficiency of β-Carotene

The encapsulation efficiency (EE) was measured using Tan’s method [20]. Approximately 1 mL of the β-carotene emulsion was placed in a centrifuge tube; 3 mL of n-hexane was added and then vortexed vigorously for 30 s. The mixture was centrifuged at 10,000× *g* for 5 min; the supernatant containing free β-carotene was collected; the unencapsulated β-carotene was extracted completely. The absorbance was measured at 450 nm for quantification, and each experiment was performed in triplicate. The total amount of β-carotene was determined according to the method in Section 2.3. The EE (%) of the emulsion to the β-carotene was calculated using the following formula:(1)$$\mathrm{EE}\left(\%\right)=\frac{\mathrm{total}\;\mathrm{amount}\;\mathrm{of}\;\beta \text{-carotene}\;-\mathrm{free}\;\mathrm{amount}\;\mathrm{of}\;\beta \text{-carotene}}{\mathrm{total}\;\mathrm{amount}\;\mathrm{of}\;\beta \text{-carotene}\;}\times 100\%$$

### 2.5. Retention Rate of β-Carotene

The newly prepared β-carotene emulsion was immediately transferred to a sealed glass bottle and then stored at different temperatures (25 °C and 37 °C) under dark conditions for 30 days. At different storage times, 1 mL of the emulsion sample was taken to determine the concentration of β-carotene in the emulsion as described in accordance with Section 2.3, and the retention rate (RR) was calculated using the following formula:(2)$$\mathrm{RR}\left(\%\right)=\frac{C_{t}}{C_{0}}\times 100\%$$
where C_t_ is the concentration of β-carotene after storage at the corresponding temperature for t days, and C_0_ is the concentration of β-carotene in the initial emulsion.

### 2.6. Construction of an In Vitro Simulated Gastrointestinal Digestion Model

The in vitro simulated gastrointestinal digestion model was used to explore the digestion behavior of the β-carotene emulsion in the gastrointestinal tract. The digestion model was constructed using the method reported by Lin [21] and Yi [22], with certain modifications. The use of pH-stat in Lin and Yi’s digestion model study to control pH during the digestive phase of the gut allowed for kinetic studies and better assessment of bioavailability. The digestion sample solution, simulated gastric fluid (SGF), and simulated intestinal fluid (SIF) were mixed in a ratio of 1:1:2. The digestion process was conducted under dark conditions, and the specific method was as follows:

#### 2.6.1. Digestion Sample Solution

Approximately 5 mL of ultrapure water was added to 5 mL of the β-carotene emulsion to prepare a digestion sample solution with a total volume of 10 mL.

#### 2.6.2. Simulated Gastric Fluid

Approximately 10 mL of SGF (containing 2 g/L of sodium chloride, 3.2 g/L of pepsin, and 7 mL/L of concentrated hydrochloric acid) was prepared. The pH was adjusted to 2.0 with 0.5 M NaOH, and the fluid was incubated at 37 °C for 5 min. The digestion sample solution was added to the SGF proportionally, and the mixture was placed in a water bath shaker (37 °C, 250 rpm) for digestion for 1 h. A sample was taken after gastric digestion was completed for subsequent analysis.

#### 2.6.3. Simulated Intestinal Fluid

Up to 20 mL of SIF (containing 8 mg/mL of bile salts, 1 mg/mL of pancreatin, and 5 mM CaCl_2_; the pH of phosphate-buffered saline was adjusted to 7.0) was prepared and incubated at 37 °C for 5 min. The SIF was added in proportion to the sample solution after gastric digestion, and the pH of the digestion solution was immediately adjusted to 7.0 with 0.2 M NaOH and then digested in a water bath shaker (37 °C, 100 rpm) for 2 h.

Fat hydrolysis occurs primarily in the small intestine. During the entire period of intestinal digestion, 0.2 M NaOH was used to maintain the pH of the digestive juice at 7.0. The volume of NaOH consumed and the time of addition were recorded. The release of free fatty acids (FFA) was calculated using the following formula [23]. In addition, samples were taken after intestinal digestion was completed for subsequent analysis.
(3)$$\mathrm{FFA}\left(\%\right)=100\%\times \left(\frac{V_{\mathrm{NaOH}}\times M_{\mathrm{NaOH}}\times {\mathrm{MW}}_{\mathrm{lipid}}}{m_{\mathrm{lipid}}\times 2}\right)$$
where V_NaOH_ is the consumption volume of NaOH (mL), M_NaOH_ is the molar concentration of NaOH (M), MW_lipid_ is the molar molecular weight of soybean oil (876 g/mol), and $m_{\mathrm{lipid}}$ is the content of soybean oil (g).

### 2.7. Microscopic Morphology of Digested Products

A sample of the digestive juice was taken after gastric digestion and intestinal digestion in accordance with the method reported by Liu [24]. Confocal laser scanning microscopy (TCS SP8, Leica Microsystems Inc., Wetzlar, Germany) was employed to observe the structural changes in the β-carotene emulsion during digestion. The fluorescent dye Nile Red (0.1%, *w*/*v*) was prepared with isopropanol. One milliliter of the digestive juice was stained with 50 μL of Nile Red; after that, it was dropped onto a glass slide and then covered with a glass slide before the observation. The excitation wavelength of Nile Red is 488 nm and its emission wavelength is 570 nm. After staining with Nile Red, the color of the soybean oil became green.

### 2.8. Bioaccessibility of β-Carotene

The bioaccessibility rate of a certain nutrient refers to the proportion of the ingested nutrient that can be provided to the body for absorption after digestion, that is, the proportion that can be absorbed by the small intestinal epithelial cells after being digested by the gastrointestinal tract. As β-Carotene is a fat-soluble biologically active substance, its bioaccessibility is the ratio of the amount of β-carotene in the micelles after digestion to the initial amount of β-carotene in the digestive system, which is calculated using the following formula. In accordance with the method reported by Qian [25], a sample was centrifuged at 15,000× *g* for 30 min at 4 °C after gastrointestinal digestion to obtain a 3-layer separation product. The upper layer consisted of undigested oil, the lower layer constituted hard-to-digest or undigested solids, and the middle layer was the micellar layer. The intermediate micellar phase was drawn with a needle tube and then filtered with a 0.22 μm filter for quantitative analysis. The total amount of β-carotene in the micelles was determined by the method used in Section 2.3.
(4)$$\mathrm{Bioaccessibility}\left(\%\right)=100\%\times \left(\frac{A_{micelle}}{A_{initial}}\right)$$
where A_micelle_ refers to the total amount of β-carotene in micelles, and A_initial_ denotes the initial amount of β-carotene in the digestive system.

### 2.9. Cytotoxicity Testing

The CCK-8 method was used to evaluate the cytotoxicity of DIPI and β-carotene micelles to mouse fibroblasts (3T3). The 3T3 cells were cultured in the DMEM medium, supplemented with 10% fetal bovine serum and 1% double antibody, and cultured in an incubator at 37 °C with 5% CO_2_. Trypsin was used to separate the cells from the culture flask. Normally cultured and stable 3T3 cells were seeded in a 96-well plate at a density of 60,000 cells/well and cultured for 24 h. Six parallel wells were set for each sample at each concentration. After 90% confluency was reached, samples with varying concentrations were added, and the culture was continued for 24 h. The relative viability of the cells was determined following the detection method specified in the CCK-8 Kit (YEASEN, China). The cells treated with the pure medium were used as a negative control. When the relative cell viability was ≥80%, the sample at this concentration was considered noncytotoxic [26,27].

## 3. Results and Discussion

### 3.1. Encapsulation Efficiency and Retention Rate of β-Carotene

The standard curve equation of β-carotene measured in this experiment is y = 0.1183x + 0.0289, and the correlation coefficient R^2^ is 0.9975, indicating a good fit.

The EEs of DIPIPE and TE emulsions to β-carotene were 97.3 ± 0.8% and 77.1 ± 1.1%, respectively, indicating that most of the β-carotene was encapsulated in the emulsion system. Compared with TE, the Pickering emulsion exerts a better encapsulation effect. As β-Carotene is relatively sensitive to temperature, the change in RR of β-carotene was evaluated over time under different storage temperatures (25 °C and 37 °C); the results are shown in Figure 1a. As shown in the figure, with the emulsion carrier remaining constant, the β-carotene content decreases more slowly under low-temperature (25 °C) than high-temperature (37 °C) storage conditions; the RR is higher. In addition, at the same storage temperature, the RR of β-carotene in TE was significantly lower than that in DIPIPE. After storage at 25 °C for 30 days, the RRs of β-carotene in DIPIPE and TE were 79.4 ± 1.2% and 46.7 ± 2.5%, respectively. After storage at 37 °C for 30 days, the RR of β-carotene in DIPIPE was 72.1 ± 2.6%, whereas that in TE was only 23.6 ± 1.4%. The appearance of the emulsion (Figure 1b) suggested that the Tween β-carotene emulsion was delaminated, whereas the DIPI-stabilized β-carotene Pickering emulsion was stable. The emulsion showed no aggregation or phase separation, and it could be inverted.

Comprehensive analysis indicates that compared with TE, DIPIPE could significantly improve the encapsulation efficiency and the retention rate of β-carotene, and it could more effectively delay its degradation and improve its chemical stability. The main reason is that previous laboratory studies proved that DIPIPE had a gel grid structure [18], and the thick interface layer of the Pickering emulsion could wrap the β-carotene oil phase in the grid structure. Consequently, the diffusion of β-carotene to the outer water phase is blocked, increasing the RR.

### 3.2. Simulated In Vitro Digestion

#### 3.2.1. Simulation of the Release of FFA in Intestinal Digestion

After being digested by the stomach, the emulsion reaches the intestinal fluid. Owing to physical impact and the influence of surface-active components such as bile salts and phospholipids, the emulsion is decomposed into an oil–water dispersion system: lipase is adsorbed on the oil–water interface, and the lipids are decomposed; and β-Carotene is released, digested, and absorbed by small intestinal epithelial cells. Therefore, studying the FFA release of emulsion bears significance for the absorption of β-carotene. Fat hydrolysis is mainly an interfacial process. Therefore, the size of the interface area of the oil droplets, in addition to its physical and chemical properties, plays a decisive role in the degree of hydrolysis.

As shown in Figure 2a, in the initial stages of digestion from 0–30 min, the FFA is rapidly released; during the digestion stage of 30–60 min, the release rate of FFA is significantly reduced; in the digestion stage of 60–120 min, FFA in TE no longer increased, and the FFA in DIPIPE exhibited a slowly increasing trend; when the digestion endpoint was reached, the FFA in TE and DIPIPE were 75.21 ± 1.6% and 47.59 ± 1.4%, respectively. Simultaneously, during the digestion stage of the small intestine, the amount of FFA released by DIPIPE was significantly less than that of TE, indicating that DIPIPE delayed fat degradation. The reason is that the interface thickness of TE was much smaller than that of the Pickering emulsion, and lipase could be easily adsorbed on the oil–water interface to decompose grease, hence the higher rate of lipolysis of TE. However, for the Pickering emulsion, the oil was wrapped by a thicker protein interface layer, and the lipase had to penetrate or diffuse to the particle interface layer before reacting with it, thereby reducing the reaction efficiency of the lipase. With the accumulation of fat hydrolysates at the interface, the rate of fat hydrolysis gradually decreased.

#### 3.2.2. Effect of In Vitro Digestion on the Microstructure of the Emulsion

After the simulated digestion stage of the gastrointestinal tract, the microstructure of the two β-carotene emulsions changed (Figure 2b). The green structure is the β-carotene oil phase. Before the simulated in vitro digestion, the droplets in the two emulsions were spherical, and the oil droplets in DIPIPE were wrapped in a network structure; these observations were consistent with the previous RR and FFA release results. After gastric digestion, TE showed single dispersed droplets without flocculation, whereas DIPIPE exhibited small droplets with a flocculated structure. These observations might be attributable to the low pH and the high ionic strength of the SGF. Protein is easily hydrolyzed by pepsin into polypeptides of different chain lengths. Polypeptides typically exhibit poor emulsification properties; consequently, emulsions often undergo flocculation and coagulation. During digestion, no aggregation was observed in TE, so the contact area was larger, which facilitated the entry of lipase into the lipid droplet surface, which resulted in a high degree of lipid hydrolysis, resulting in the release of more β-carotene. After the digestion of the small intestine, several tiny droplets were found in TE, indicating that a large amount of lipids was hydrolyzed; meanwhile, DIPIPE still contained a small number of flocculated structures, which reduced the chance of contact between lipase and the lipid core, thus delaying fat hydrolysis. The aforementioned experimental phenomenon was consistent with the release of FFA, indicating that the microstructure of the emulsion directly affected fat hydrolysis during digestion and β-carotene release.

#### 3.2.3. Bioaccessibility of β-Carotene

Oil hydrolysis is accompanied by β-carotene release. After the emulsion is digested by the simulated stomach in vitro, lipid hydrolysates such as diacylglycerol, monoglycerol, and FFA are released from the emulsion droplet interface and form micelles with bile salts and phospholipids. The β-carotene initially dissolved in the oil is eventually dissolved in the micelles (Figure 3a) and it becomes bioavailable. The bioaccessibility of β-carotene is shown in Figure 3b. The bioaccessibility of β-carotene is 62.4 ± 0.80% in TE and 37.1 ± 0.77% in DIPIPE, exceeding that in pure soybean oil (approximately 21%) [28]. The degree of emulsion fat hydrolysis is found to be related to micelle formation; the higher the FFA release rate, the higher the degree of fat hydrolysis, and the easier the formation of micelles. That is, the degree of lipolysis is positively correlated with β-carotene bioavailability [29,30]. Therefore, DIPIPE exhibits a lower bioaccessibility than TE. However, Tween is a synthetic emulsifier with potential safety issues [31], and it cannot provide long-term stability to the emulsion at room temperature and higher temperatures. Plant protein is a natural emulsifier with reliable levels of safety and biocompatibility, and it can produce polypeptide fragments after protease hydrolysis. These polypeptides may contain antioxidants, lower blood pressure, reduce blood sugar, and exert other effects. Compared with the results in the literature, the bioaccessibility of β-carotene in DIPIPE is similar to that in the β-carotene Pickering emulsion prepared by Fu et al. [5] using a wheat gluten-xanthan gum complex; however, these results were higher than the β-carotene emulsion prepared by Liang et al. [32] using modified starch. In general, DIPIPE is a good carrier for β-carotene.

#### 3.2.4. Cytotoxicity Testing

The results of 3T3 cytotoxicity detection at different concentrations of DIPI are shown in Figure 4a. Compared with that of the control group, the concentration of the DIPI group was in the 10–90 μg/mL range, and the relative cell viability exceeded 94%, indicating that DIPI exhibited no cytotoxicity within this concentration range. Therefore, DIPI has no potential toxicity and it can be used as a food-grade Pickering stabilizer. The O/W-type Pickering emulsion stabilized by DIPI is a food-grade emulsion, and it is an excellent carrier for embedding and delivering β-carotene.

The CCK-8 method was used to determine the potential toxicity of micelle samples collected after simulated digestion to 3T3 cells. For TE (Figure 4b), the β-carotene in chylomicrons was not cytotoxic in the concentration range of 0.87–7.80 μg/mL, and it can promote the viability of 3T3 cells. The concentration of β-carotene was dose-dependent with cell viability, which may be the active function of β-carotene to promote cell proliferation. In addition, there was no thick interface layer in TE, and the reaction degree of active substances such as bile salts and enzymes was high during the simulated digestion process. Therefore, the content of bile salts in the micelle was low, and it would not damage cells. For DIPIPE (Figure 4c), low concentrations (0.72–2.16 μg/mL) of β-carotene in chylomicrons had a promoting effect on the viability of 3T3 cells, and there was a slight inhibitory effect when the concentration was increased. However, the cell viability was still >80%, and therefore non-cytotoxic. The reason for the decreasing trend of cell viability in Figure 4c may be that the increased content of bile salts in the micelles caused slight damage to the cells [33]. However, in the literature, the mucosal layer in the intestine can be used as a physical protective barrier during actual digestion to prevent harmful substances such as bile salts from damaging or coming into contact with cells [34,35]. Calcium ions and coenzyme factors secreted by the human body can effectively reduce the damage of bile salt molecules to the small intestinal epithelial cell membrane [36].

## 4. Conclusions

In this study, β-Carotene was successfully encapsulated in a Pickering emulsion stabilized by DIPI. DIPIPE is a viscoelastic emulsion that has a higher encapsulation rate, retention rate, and storage stability for β-carotene, compared with TE. In vitro simulated digestion studies have shown that DIPIPE has a gel network structure and a thicker interface layer, which can delay β-carotene release, and it thus has a lower bioaccessibility than that of the control group. Cytotoxicity experiments indicate that DIPI and β-carotene micelles exert no toxic effects on 3T3 cells within a specific concentration range. Comprehensive analysis shows that DIPIPE is a good carrier for embedding and transporting active substances with certain safety. This study provides a theoretical basis for the application of an emulsion delivery system, and it provides a simple and effective technical approach to solve the defects of β-carotene application in functional food. The lack of characterization (stability, size, and PDI) of Pickering and Tween emulsions is seen as a limitation of this study, as they are the main factors affecting bioaccessibility and cytotoxicity. This is a new point of discussion, and in the following studies we will explore the effects of emulsion properties on bioaccessibility and cytotoxicity.

## Figures and Tables

**Figure 1 foods-11-00936-f001:**
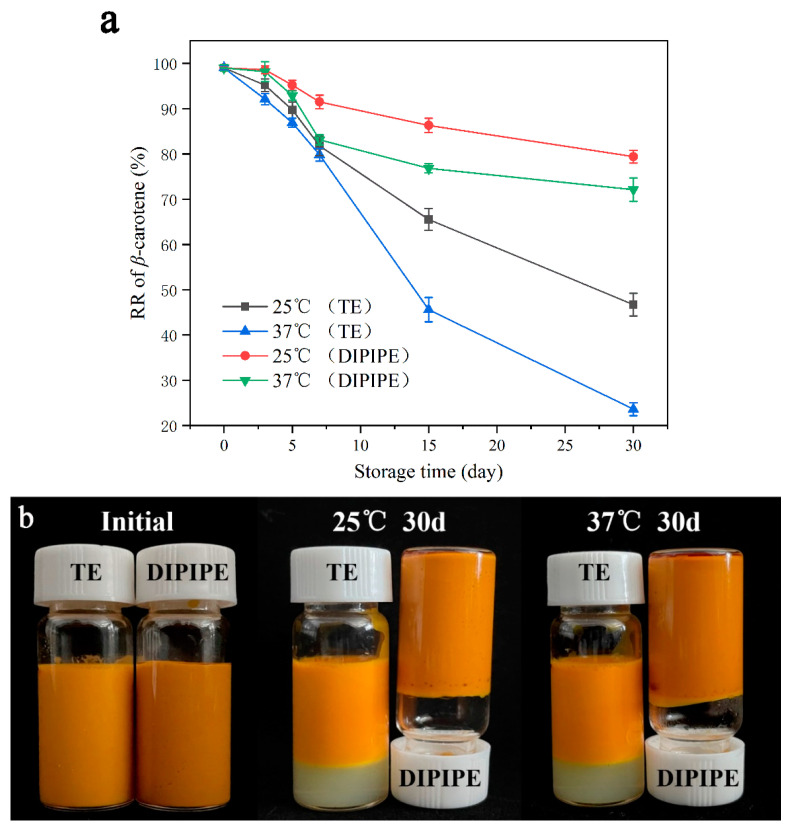
(**a**) Changes in β-carotene retention rate over time at different storage temperatures. (**b**) Appearance of the β-carotene emulsion after storage for 30 days at 25 °C and 37 °C.

**Figure 2 foods-11-00936-f002:**
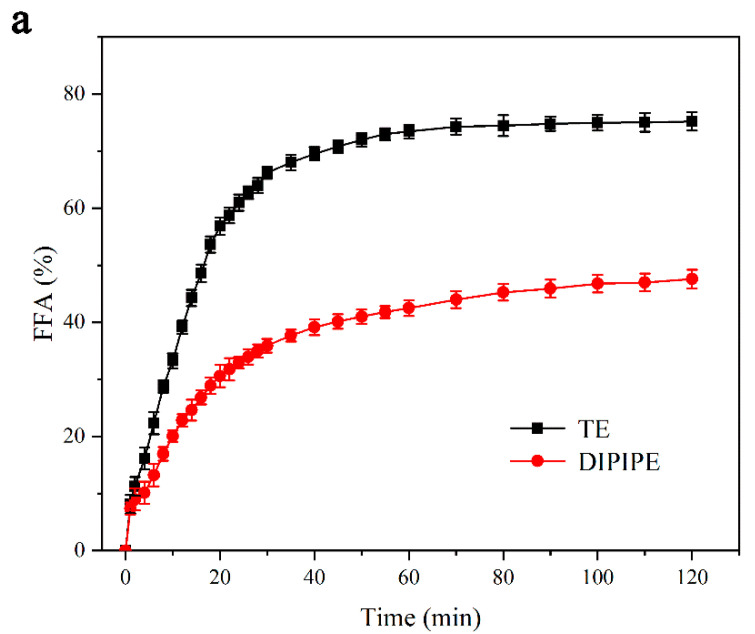
(**a**) Changes in the free fatty acid of β-carotene emulsion over time during simulated intestinal digestion. (**b**) Microstructure of the emulsion during in vitro simulated digestion (scale of initial emulsion = 100 μm; scale of stomach and small intestine digestive juice = 50 μm).

**Figure 3 foods-11-00936-f003:**
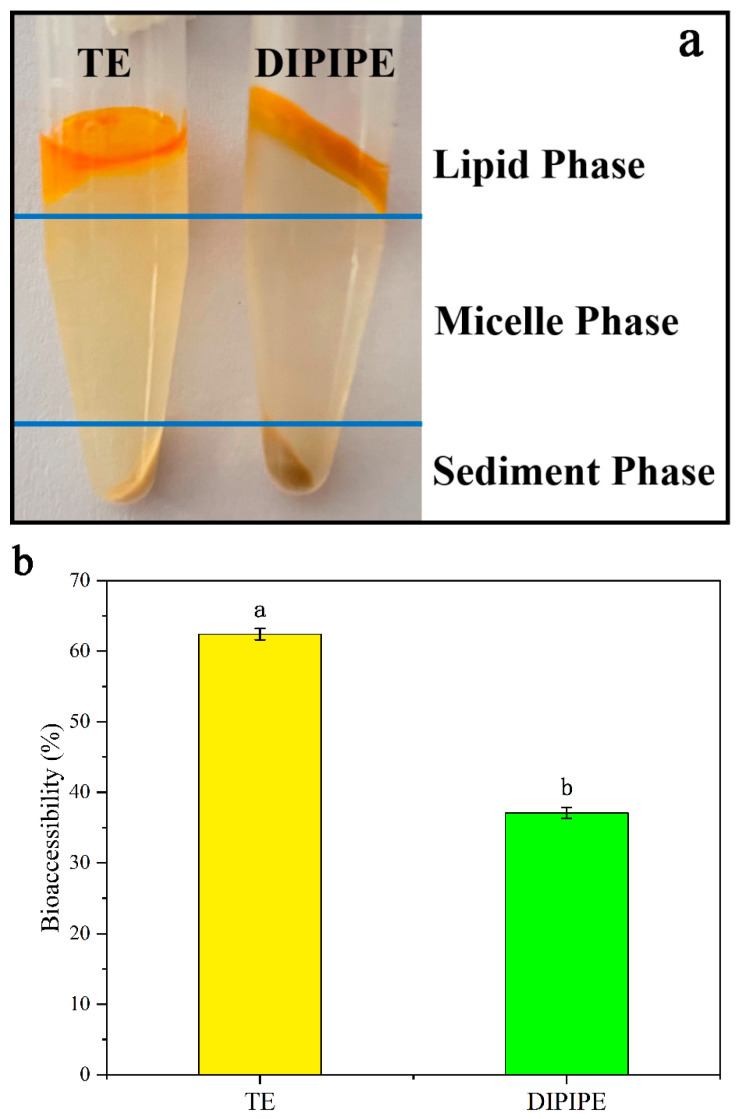
(**a**) Appearance of the micellar layer. (**b**) Bioaccessibility of β-carotene (TE and DIPIPE data results are marked with different lowercase letters indicating that the difference is significant at the 0.05 level).

**Figure 4 foods-11-00936-f004:**
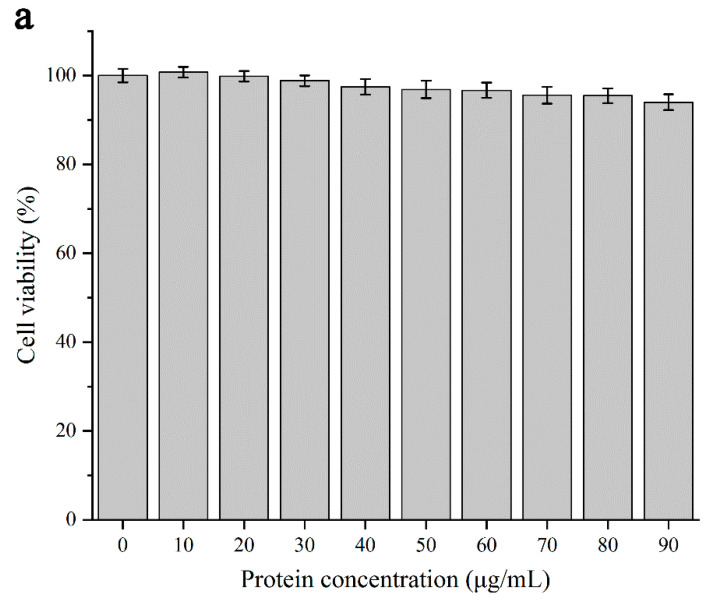
(**a**) Cytotoxic effects of DIPI on 3T3 cells. (**b**)Toxic effects of β-carotene-containing micelles (TE) on cells. (**c**) Toxic effects of β-carotene-containing micelles (DIPIPE) on cells.

## Data Availability

Data is contained within the article.

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
