# Peer review of "Encapsulation of a Desmodium intortum Protein Isolate Pickering Emulsion of β-Carotene: Stability, Bioaccesibility and Cytotoxicity"

_foods, 2022, doi:10.3390/foods11070936_

Round 1
Reviewer 1 Report
The authors significantly improved their manuscript. However, they did not answer to all my questions. Why did the authors choose very high temperature (90 oC) for beta-carotene dissolution (line 93). Moreover, more detailed discussions and comments should be presented about Figure 4b.
Author Response
Comment 1--- The authors significantly improved their manuscript. However, they did not answer to all my questions. Why did the authors choose very high temperature (90oC) for beta-carotene dissolution (line 93). Moreover, more detailed discussions and comments should be presented about Figure 4b.
Response 1: In this study, the melting point of β-carotene crystal powder was 183°C, which was only slightly soluble in oil, and it was very difficult to directly dissolve the crystal powder in oil at room temperature. The original process of dissolving β-carotene in vegetable oil needs to be dissolved at 150°C for a long time. In this study, in order to avoid thermal degradation of β-carotene, ultrasonic pretreatment was used and then heated at 90°C to dissolve β-carotene. Additionally, we have discussed and commented in more detail on Figure 4b.
Reviewer 2 Report
Dear Authors,
Thank you for the opportunity to review the paper entitled "Encapsulation of a Desmodium intortum protein isolate Pickering emulsion of β-carotene: stability, bioaccesibility and cytotoxicity". I think that this paper is improved.
Prior to publishing, please adopt some additional suggestions. Some of my overall comments for this paper:
-Editing English is required throughout the manuscript. The paper is very difficult to read, due to insufficient knowledge of English.
- line 13 “owing” is not bold
- line 51-67 – highlight in better manner the significant of β-carotene
line 269 in vitro write italic
- in conclusion, part add carefully that given the topicality of food industry waste utilization and isolation of β-carotene from such sources, this paper is not acceptable from the point of view of environmental protection, but is promising….add all advantages of study…
lines 73-80 This is not the goal of the paper. Some facts from the previous research are given, which is not adequate to this paper. Improve this part again.
Figure 1,2 and 3 do not have a) and b) on figure, while in figure caption is stand a) and b). Please add on figure letters
Figure 4 does not have a), b) and c) on figure, while in figure caption is stand a) and b). Please add on figure letters
Author Response
Comment 1--- -Editing English is required throughout the manuscript. The paper is very difficult to read, due to insufficient knowledge of English.
Response 1: We have revised the manuscript.
Comment 2--- - line 13 “owing” is not bold
Response 2: We have revised the manuscript.
Comment 3--- - line 51-67 – highlight in better manner the significant of β-carotene
Response 3: We revised this content according to the suggestions of reviewer 2 (see the amended manuscript).
Comment 4--- line 269 in vitro write italic
Response 4: We have revised the manuscript.
Comment 5--- - in conclusion, part add carefully that given the topicality of food industry waste utilization and isolation of β-carotene from such sources, this paper is not acceptable from the point of view of environmental protection, but is promising….add all advantages of study…
Response 5: We added this content according to the suggestions of reviewer 2 (see the amended manuscript).
Comment 6--- lines 73-80 This is not the goal of the paper. Some facts from the previous research are given, which is not adequate to this paper. Improve this part again.
Response 6: We have improved added this content.
Comment 7--- Figure 1,2 and 3 do not have a) and b) on figure, while in figure caption is stand a) and b). Please add on figure letters.
Response 7: We have added letters to Figure 1, 2 and 3.
Comment 8--- Figure 4 does not have a), b) and c) on figure, while in figure caption is stand a) and b). Please add on figure letters
Response 8: We have added letters to Figure 4.
Reviewer 3 Report
Comments that have not been answered:
Why was a Lin and Yi digestion model used, when in fact an international consensus has been reached (known as the INFOGEST model) for suitable and comparable digestion models? The authors should specify why the INFOGEST model was not applied, and how and why their model differs the INFOGEST model.
The activity of pancreatin is not reported. Without activity of pancreatin the results are not reproducible.
Figure 2 a and Figure 4a are not required. It is quite easy to visualize 2 numbers.
The characterization of both Pickering and Tween emulsions is missing i.e. stability, size, and PDI.
Author Response
Comment 1--- Why was a Lin and Yi digestion model used, when in fact an international consensus has been reached (known as the INFOGEST model) for suitable and comparable digestion models? The authors should specify why the INFOGEST model was not applied, and how and why their model differs the INFOGEST model.
Response 1: The differences between Lin and Yi’s digestion model and the INFOGEST model can be provided as follows. First, the steps involved in the digestion process are different, and the INFOGEST model includes the oral digestion stage. Second, the chemical composition and ratio of each digestion stage are different. Third, the mechanical stress during digestion is different. The INFOGEST model is an in vitro static digestion model. Each digestion stage is carried out at a constant pH level and enzyme concentration. The model is simple and easy to operate and can effectively predict the results. However, the INFOGEST model also has some limitations, such as simplifying the physiological process and failing to mimic the dynamic aspects of the digestive process. The use of pH-stat in the digestion model study of Lin and Yi to control the pH value during the intestinal digestion stage enables kinetic studies and a better assessment of bioavailability. Therefore, this digestion model was used.
Comment 2--- The activity of pancreatin is not reported. Without activity of pancreatin the results are not reproducible.
Response 2: A report of the pancreatin used for research is provided below. We have supplemented this information in the manuscript.
Comment 3--- Figure 2 a and Figure 4a are not required. It is quite easy to visualize 2 numbers.
Response 3: Regarding Figures 2a and Figures 4a should be deleted , your suggestion is very pertinent and beneficial. However, considering the sufficiency of the experimental results, we only delete Figure 2a.
Comment 4--- The characterization of both Pickering and Tween emulsions is missing i.e. stability, size, and PDI.
Response 4: We want to thank the reviewers for their constructive comments and suggestions. We understand that the characterization of the relevant properties of Pickering and Tween emulsions can better explain the conclusions of this study. However, in the present study, our main concern is the encapsulation properties of the emulsions for carotene. Therefore, the stability, size, and PDI of the emulsions were not investigated. In our following research, we will further improve the related investigation of emulsions.

Round 2
Reviewer 1 Report
The manuscript is ready for publication. The authors addressed to all my comments.
Author Response
Thank you for your help in our manuscript.
Reviewer 3 Report
In the response, the authors state that emulsion characterization is not part of the current research. I disagree. Since the authors compare 2 emulsions - made with Tween and with protein isolate, the factors which influence bioaccessibility and cytotoxicity are all dependent on the particle size of the emulsion.
Was the enzyme activity measured by the researchers, or reported from the manufacturer (if reported - a reference should be provided)? It is not clear. Again - an activity range does not allow for assessing the replicability of the results.
Author Response
Comment 1--- In the response, the authors state that emulsion characterization is not part of the current research. I disagree. Since the authors compare 2 emulsions - made with Tween and with protein isolate, the factors which influence bioaccessibility and cytotoxicity are all dependent on the particle size of the emulsion.
Response 1: Thank you for your comment, your suggestion is very pertinent and beneficial. We understand that the characterization of the relevant properties of Pickering and Tween emulsions can better explain the experimental results. In this study, we mainly focused on the results of emulsion bioaccessibility and cytotoxicity, so the factors influencing the results were not explored. The effect of emulsion particle size on bioaccessibility and cytotoxicity is a new point of discussion, and in the following studies, we will further refine the related experiments of emulsions.
Comment 2--- - Was the enzyme activity measured by the researchers, or reported from the manufacturer (if reported - a reference should be provided)? It is not clear. Again - an activity range does not allow for assessing the replicability of the results.
Response 2: The enzyme activity is reported from the manufacturer and the product information report is as follows:

This manuscript is a resubmission of an earlier submission. The following is a list of the peer review reports and author responses from that submission.
Round 1
Reviewer 1 Report
The present work presents an intresting approach to encapsulate the valuable carotenoid beta-carotene in order to increase its stability and bioaccessibility through a Pickering emulsion.
The manuscript needs some improvement:
-Editing and English improvement is required throughout the manuscript. More specifically:
- Line 15: "Owing..." should not be bold
- The sentence in lines 44-46 needs rephrasing
- Lines 51, 52, 83, 89, 103, 111, 119, 128, 174 and wherever else mentioned in the manuscript "β-carotene .." in the beginng of a sentence should be "β-Carotene ...
- Line 116: "lost its colour..." could be replaced by "became colorless"
- Line 117: "...were combined and then dilute..." should be corrected to "...were combined and then diluted..."
- Line 152: "... after intestinal digestion..." should be corrected to "...after gastric digestion..."
-Line 68: If not mistaken, the authors are refering to a previous work of theirs. Which one is this? Is it published or they refer to unpublished data? This should be clarified and if the work is published, it should be cited in the text and added in the reference list.
-Figure 2a should be deleted as the results of the encapsulating efficiency are given in the text in line 214.
-All figures should have indications to clearly show which one is the (a), the (b) etc.
-Lines 265, 395, 405, : "in vitro..." should be italics
-Lines 406, 430 the year of publication should be bold and the volume in italics
Reviewer 2 Report
The present study is interesting, but it should be greatly improved, specially in the justification and discussion:
TITLE: “Encapsulation of a Desmodium intortum protein isolate Pickering emulsion of β-carotene: stability, bioaccesibility and cytotoxicity” better than “Encapsulation properties of a Desmodium intortum protein isolate Pickering emulsion on β-carotene: Stability and digestive characteristics”
KEY WORDS: “cytotoxicity” should be included.
INTRODUCTION: in lines 72 authors have indicated the use of Tween 20 emulsion as a traditional way to encapsulate β-carotene, while other strategies have been shown in lines 62-67. These aspects should be clarified to justify the use of Tween 20 in the present study. In the same way, the cytotoxicity studies should be also justified (why is this determination needed?) as well as the use of DIPI.
Line 71: “The DIPI Pickering emulsion (DIPIPE) obtained in the previous study”. A reference is needed.
Lines 75-78: these lines seem to be results from the study, please, delete them.
MATERIAL AND METHODS
Line 81 “DIPI is made by the laboratory”. This issue should be explained and referenced. How did you obtain DIPI? When did you obtain Desmodium intortum? A detailed explanation of these aspects is required.
2.2. Preparation of Emulsion-encapsulated β-Carotene: after reading this section, I noticed that the comparison of a real emulsion, with Tween, with a Pickering emulsion, with a solid wall, is not so appropriate.
Line 105: “at a certain concentration”. The ranges of the calibration curve should be shown.
2.4. Encapsulation Efficiency of β-carotene: authors should use the total amount of β-carotene determined as explained in point 2.3 better than the total amount of β-carotene added to calculate the EE.
2.5. Retention Rate of β-carotene: what about the stability of the emulsions during the storage? I miss a stability measure, such as creaming index (CI). This is crucial.
2.6.3. Simulated Intestinal Fluid: why did author calculate the percentage of FFA?
Lines 197-198: “When the relative cell viability was ≥80%, the sample at this concentration was considered noncytotoxic” A reference is needed.
I miss the statistical design, the number of emulsions, replicates… This is also crucial for this kind of studies.
RESULTS AND DISCUSSION
3.1. Evaluation of DIPI Cytotoxicity: it is surprising to find this subsection since it has not been explained in the material and method section. Besides, it should be included in the other section on cytotoxicity and discuss all the results.
In general, authors should improve the discussion about the differences between the two compared emulsions.
Reviewer 3 Report
Dear Editor,
Thank you for the opportunity to review the paper entitled "Encapsulation properties of a Desmodium intortum protein isolate Pickering emulsion on β-carotene: Stability and digestive characteristics ". I think that this paper must be improved before any further steps in publishing process.
Some of my overall comments for this paper:
The biggest objection is the inconsistency, as well as the lack of a certain concept that would explain what has been done in previous research, and what specifically in this. Given the topicality of food industry waste utilization and isolation of β-carotene from such sources, this paper is not acceptable from the point of view of environmental protection. This kind of work would make sense if the used β-carotene represents a waste stream and is successfully used in this way. The paper is very difficult to read, due to insufficient knowledge of English.
Materials and methods did not provide enough information about concept, but also each paragraph is lack of information about used methodology. The used methods are good, but nothing more than that.
Abstract is not enough attractive to readers, some crucial information about research is missing.
Please, try to avoid repetition of same words in title and keywords.
Authors did not approached importance of Desmodium intortum as well as used bioactive compounds.
lines 68-78 This is not the goal of the paper, on the contrary. Some facts from the previous research are given, which is not adequate to this paper. Also, using "we" speaks enough about the lack of knowledge of the English language.
The real discussion and conclusion, as well as presentation of advantages and disadvantages are missed in this paper.
Reviewer 4 Report
The authors describe the preparation of beta-carotene loaded Pickering emulsion stabilized by Desmodium intortum protein isolate. The authors conclude that Desmodium intortum protein isolate can be used for the delivery systems of beta-carotene and have the advantages compared to Tween 20. The manuscript has to be improved.
- Line 71. What does 0.7 mean?
- Lines 68-71. The reference should be provided.
- For readers, more information on Desmodium intortum protein isolate should be presented in the Introduction.
- Line 90. The temperature of 90 oC is very high. Could the authors estimate what amount of beta-carotene was degraded.
- Line 184. How was the amount of beta-carotene in micelles calculated?
- Figure 1 and Figure 5. What was statistical analysis done. How many times were the experiments repeated. The section Materials and Methods lacks the description of statistical analysis.
- Lines 235-236. The reference has to be provided.
Reviewer 5 Report
Line 40 : “It has become one of the main objects of food functional 40 factor carrier research [4]” is not clear what it refers to, and does not actually add anything new.
Line 61 Relevant research shows that β-carotene can be embedded in 61 emulsions, liposomes, solid lipid nanoparticles, and macromolecular complexes [15,16]. So how is the Pickering emulsion better than the previous methods of embedding?
Why specifically was Desmodium intortum used for Pickering emulsions?
If the isolate is in contact with water, can it really be classified as a Pickering emulsion?
Why was a Lin and Yi digestion model used, when in fact an international consensus has been reached (known as the INFOGEST model) for suitable and comparable digestion models? The authors should specify why the INFOGEST model was not applied, and how and why their model differs the INFOGEST model. Controls of the experiment are not shown in Figure 3a.
The activity of pancreatin is not reported.
Figure 2 a and Figure 4a are not required. It is quite easy to visualize 2 numbers.
The characterization of both Pickering and Tween emulsions is missing i.e. stability, size, and PDI.